# Understanding COVID-19 Vaccine Hesitancy among Healthcare Workers in South Africa

**DOI:** 10.3390/vaccines11020414

**Published:** 2023-02-10

**Authors:** Gavin George, Phiwe Babalo Nota, Michael Strauss, Emma Lansdell, Remco Peters, Petra Brysiewicz, Nisha Nadesan-Reddy, Douglas Wassenaar

**Affiliations:** 1Health Economics and HIV and AIDS Research Division (HEARD), University of KwaZulu-Natal, Durban 4041, South Africa; 2Division of Social Medicine and Global Health, Lund University, 223 63 Lund, Sweden; 3Research Unit, Foundation for Professional Development, East London 5241, South Africa; 4School of Nursing & Public Health, University of KwaZulu-Natal, Durban 4041, South Africa; 5South African Research Ethics Training Initiative, School of Applied Human Sciences, University of KwaZulu-Natal, Pietermaritzburg 3209, South Africa

**Keywords:** healthcare workers, vaccine hesitancy, South Africa, COVID-19

## Abstract

Healthcare workers (HCWs) were the first population group offered coronavirus disease 2019 (COVID-19) vaccines in South Africa because they were considered to be at higher risk of infection and required protecting as they were a critical resource to the health system. In some contexts, vaccine uptake among HCWs has been slow, with several studies citing persistent concerns about vaccine safety and effectiveness. This study aimed to determine vaccine uptake among HCWs in South Africa whilst identifying what drives vaccine hesitancy among HCWs. We adopted a multimethod approach, utilising both a survey and in-depth interviews amongst a sample of HCWs in South Africa. In a sample of 7763 HCWS, 89% were vaccinated, with hesitancy highest among younger HCWs, males, and those working in the private sector. Among those who were hesitant, consistent with the literature, HCWs raised concerns about the safety and effectiveness of the vaccine. Examining this further, our data revealed that safety and effectiveness concerns were formed due to first-hand witnessing of patients presenting with side-effects, concern over perceived lack of scientific rigor in developing the vaccine, confidence in the body’s immune system to stave off serious illness, and both a general lack of information and distrust in the available sources of information. This study, through discursive narratives, provides evidence elucidating what drives safety and effectiveness concerns raised by HCWs. These concerns will need to be addressed if HCWs are to effectively communicate and influence public behaviour. HCWs are key role players in the national COVID-19 vaccination programme, making it critical for this workforce to be well trained, knowledgeable, and confident if they are going to improve the uptake of vaccines among the general population in South Africa, which currently remains suboptimal.

## 1. Introduction

Healthcare workers (HCWs) have been and remain at the coalface of the coronavirus disease 2019 (COVID-19) pandemic. In recognition, there were calls for HCWs to be prioritised when the COVID-19 vaccines were initially made available [1]. This was the case in South Africa, when on 17 February 2021, the country’s first HCW was vaccinated (through an implementation study), with the target of reaching all of the estimated 1.25 million HCWs [2]. HCWs are a critical resource to protect from COVID-19 infection, as they are a vital component of the health system and play a key role in the success of a vaccination programme targeting the public [3,4,5,6,7]. Low vaccination rates among HCWs and high levels of vaccine hesitancy can have a ripple effect, resulting in decreased vaccination uptake by those who engage with HCWs clinically and personally [8].

Most research on COVID-19 vaccine hesitancy among HCWs has been conducted in high-income countries [9,10,11,12,13,14]. Studies in Africa, many of which were undertaken prior to the availability of vaccines, revealed mixed responses to the impending availability of vaccines. A study conducted in the Democratic Republic of Congo revealed that only 27.7% of HCWs in the study would accept a COVID-19 vaccine once available [15]. Another study in Ethiopia indicated that 60.3% of the sample of HCWs were hesitant to receive the COVID-19 vaccine, with HCWs below the age of 30 approximately five times more likely to be vaccine-hesitant [16]. Other studies undertaken in Africa have, however, revealed higher levels of acceptance [17,18], including research in South Africa which found that the majority (59%) of HCWs would accept the vaccine [19]. 

Factors associated with HCWs’ COVID-19 vaccine hesitancy vary across contexts and studies. Africa on the whole, in comparison to many other continents, has experienced lower rates of infection and morbidity [20], attributed largely to the continent’s relatively young and rural population, less international travel, and the early introduction of nonpharmaceutical interventions [21]. As a result, perception of risk may appear lower, resulting in low and slow uptake of vaccines [22]. Safety and efficacy concerns remain pervasive across numerous studies and remain a key factor driving hesitancy, with HCWs concerned by the perceived speed at which COVID-19 vaccines were developed [6]. In several studies, HCWs preferred to wait to review further data to see how vaccines affected others, highlighting the need for more information about the safety and effectiveness of vaccines [23,24,25,26]. These concerns persisted despite assurances from public health experts [18,24,27]. 

Given that vaccines are now globally available, it remains key to determine whether vaccine acceptance has translated into uptake. Given the role of HCWs in influencing public confidence in vaccines, there is an urgent need for data on vaccine uptake and the prevalent concerns of HCWs in Africa [28]. Approximately 25.6% of the general population in Africa has been fully vaccinated [29], including 51% in South Africa as of 12 January 2023 [30], suggesting that the drivers of hesitancy require further examination. Key to elucidating what underlying concerns may be facilitating vaccine hesitancy, is the engagement with HCWs who are tasked with the administration of the vaccine and, more importantly, remain responsible for allaying public fears and advocating for vaccine uptake. In this paper, we quantitatively examine sociodemographic and health factors associated with vaccine uptake among HCWs across South Africa. We then present the results of a qualitative exploration of the drivers of hesitancy, with the aim of identifying and exploring concerns raised by HCWs within this context, examining how these concerns have been formulated. This study is expected to contribute to the design and implementation of interventions aimed at ensuring that HCWs operate as effective vaccine advocates in the interests of personal and public health promotion. 

## 2. Materials and Methods

### 2.1. Study Design and Setting

A self-administered online survey included sociodemographic information, COVID-19 history, chronic conditions, and questions on vaccination behaviour. Survey questions were derived from a review of studies evaluating HCW hesitancy towards COVID-19 vaccines [31]. Participants interested in participating in in-depth interviews (IDIs) were asked to provide their contact details following completion of the online survey. Participants who provided their contact details were organised into two groups: vaccinated and unvaccinated. Participants from these groups were then randomly selected using the *randbetween* formula in Microsoft Excel for IDIs. The findings from the survey data and qualitative interviews were integrated when interpreting the results, and the findings were triangulated in the discussion with the extant literature. Ethics approval was obtained from the UKZN BREC (approval number BREC/00003970/2022).

### 2.2. Recruitment and Data Collection

Data collection occurred between August 2022 and October 2022. Participants were offered an incentive in the form of entry into a draw for one of ten ZAR500 (~33 USD) cash vouchers; the draw was not linked to participants’ survey responses. Consent was indicated by participants reading the consent form on the opening page of the survey and clicking ‘agree’, which enabled them to begin answering the survey questions. The Foundation for Professional Development’s (FPD) database, which comprised contact details of 88,000 HCWs at the commencement of the study, was used to recruit HCWs for this study. FPD is a private higher-educational institution that provides training to HCWs, and, with permission, records their details in the database. Randomly selected participants who indicated in the survey that they would be willing to be contacted for a follow-up interview, were recruited and scheduled for an IDI on Zoom. The online interviews lasted approximately 30 min and were recorded and transcribed.

### 2.3. Sample Size

The final analysis was undertaken on 7763 completed surveys after discarding 1488 (16%) records due to incomplete data or duplication. Sociodemographic variables were selected on the basis of prior studies of COVID-19 vaccine acceptance and uptake [28,32]. HCWs were split into three groups: nurses, doctors, and other HCWs (other healthcare workers in the study included allied health professionals (8.88%), dentists (8.60%), ambulance staff/paramedics (4.69%), dental hygienists (2.84%), and pharmacists (1.73%)). A total of 30 HCWs were recruited for IDIs. Of these, 10 were vaccinated and 20 were not vaccinated at the time of the interviews. 

### 2.4. Data Analysis

Descriptive analysis included overall frequency and percentages of independent variables, as well as frequency and percentage split by the dependent variable—vaccination status. Independent variables included age, gender, race, religion, nationality, occupation, health sector (private, public, nongovernmental organisation, and other), years worked, chronic medical conditions, and COVID-19 history. Bivariate analysis was undertaken on all variables, using logistic regression models. Statistical analyses were performed using STATA v17. Results are presented as odds ratios (ORs) with 95% confidence intervals. A *p*-value below 0.05 was considered significant. 

Qualitative data were analysed thematically using an inductive approach as prescribed by Braun and Clarke [33] in a six-step process. This process entailed the researchers (P.N. and G.G.) familiarising themselves with the data by taking notes while conducting the interviews, listening to the recoded interviews, and reading the transcripts. Zoom transcribing software was used to transcribe the recorded interviews. Transcriptions were read and codes were developed, after which themes were generated.

## 3. Results

Most participants in the sample were aged 35–49 years (40.95%) and female (69.34%). Most identified as Black African (58.33%) and South African (89.00%). The dominant religion was Christianity (82.68%). Most of the sample comprised nurses (44.45%), followed by doctors (20.42%). Most had worked as healthcare professionals for 10 or more years (63%). Participants were evenly split across the public (40.72%) and private (35.81%) sectors. More than half the sample had one or more chronic conditions (59.68%). About half (50.46%) of the sample had previously knowingly been infected with COVID-19. Across the sample of 7763 HCWS, 89% were vaccinated.

Table 1 shows the characteristics of HCWs according to vaccination status. Older participants (OR = 1.31, 95% CI: 1.07–1.60) and females (OR = 1.22, 95% CI: 1.04–1.44) were more likely to be vaccinated. Participants who listed their religion as African Spirituality or other religious beliefs, not specified here, were less likely to be vaccinated (OR = 0.52, 95% CI: 0.35–0.76) compared to Christians (OR = 0.93, 95% CI: 0.69–1.27). Doctors (OR = 0.76, 95% CI: 0.59–0.98) and other HCWs (OR = 0.63, 95% CI: 0.51–0.78) were less likely to be vaccinated than nurses. HCWs in the private sector (OR = 0.78, 95% CI: 0.63–0.96) and those working in both the public and the private sectors (OR = 0.93, 95% CI: 0.66–1.30) were less likely to be vaccinated than those working in the public sector. HCWs who had a self-reported chronic condition were more likely to vaccinate than those not reporting any chronic conditions (OR = 1.43, 95% CI: 1.20–1.69). 

Qualitative analysis focused on uncovering reasons for vaccine hesitancy amongst HCWs. Safety and the vaccine’s perceived effectiveness emerged as the dominant concerns of HCWs. However, five themes emerged from participants who were not vaccinated, providing insights into the formulation of concerns around safety and effectiveness. These included (1) *the perceived speed at which COVID-19 vaccines were developed*, (2) *observing patients presenting with apparent side-effects following vaccination*, (3) *confidence in natural immunity*, (4) *lack of trust in the sources of information, and* (5) *insufficient and misinformation about COVID-19 vaccines.* Each of these is discussed more fully below.

### 3.1. The Perceived Speed at Which COVID-19 Vaccines Were Developed 

Hesitant participants expressed concern about the perceived speed in which clinical trials for COVID-19 vaccines were undertaken and the rapid vaccine development process. Participants described how, in their own experiences in the medical field, vaccine development required long-term clinical trials to ensure safety standards were met. 


*First of all, I worked for 4 years on HIV vaccine research at the University of Cape Town in the Institute of Infectious Disease and Molecular Medicine. So, I know the amount of work that goes into creating a vaccine.*
(P25, Other HCW, M, Unvaccinated)

The account below describes how some HCWs questioned the scientific rigor in the COVID-19 vaccine development because of the perceived speed in which the vaccines became available. 


*Any other medicine trial, anything else was 5 years plus before it ever got released, and how many medications have been recorded, even after all these test trials with much fewer side-effects than we’ve seen after the vaccine. And it just gets ignored, and that got all my alarm lights up that I said that this is not sensible, it’s not scientific.*
(P26, Doctor, F, Unvaccinated)

While most vaccinated participants highlighted trust in the science and research process in developing the COVID-19 vaccines, some described how they were initially concerned about the lack of long-term clinical trial data. These concerns were rooted in the fact that lack of long-term follow-up data may result in unobserved adverse side-effects among different population groups. The account below was from a medical doctor who was vaccinated but described how she too was uncomfortable with the vaccination development process. 


*My major concerns I think, regarding the vaccine were most likely how […] fast it was released, and I know that it was sort of zipped along in the fastest possible channels. And yes, they’re trying to make it safe but actually we don’t have any proper safety data, let’s be honest because we haven’t done human studies […] The early data seemed good, but the safety of it was definitely a query for me because of how, sort of rushed along the whole process was. It was a bit uncomfortable.*
(P19, Doctor, F, Vaccinated)

### 3.2. Observing Patients Present with Side-Effects following Vaccination 

Participants were concerned about COVID-19 vaccine side-effects. Most of the participants who reported this concern gave an account of patients who had reportedly experienced adverse side-effects post vaccination.

The participant below lists some of the adverse side-effects patients had reported. 


*I had several myocardial infarction[s], several strokes, myocarditis, pericarditis, and all sorts of other problems. Flare-ups of herpes zoster, flare-up of TB, diabetes, hypertension getting out of control, and you know when I saw that trend, I started recording all the patients coming to me and asked them if they’ve been vaccinated.*
(P8, Doctor, M, Unvaccinated)

Participants frequently reported patients vaccinating and showing an immediate adverse side-effect. The account below reports several people collapsing after vaccinating. 


*There were instances whereby immediately after being injected with the vaccine other individuals collapsed almost immediately. Like before they could reach the gate leaving the hall where the vaccines were conducted, and they had to rush them to the hospital. And then with the duration of time also, we see others actually experienced partial paralysis.*
(P2, Nurse, M, Unvaccinated)

The quote below is a report from a participant who claimed that a seemingly healthy young male patient died shortly after receiving the COVID-19 vaccine. 


*There was a client, in fact, a few clients of mine, that I was doing home visits. I remember this, the young boy was 30 something, he was healthy. Him and his father went for a vaccine and 3 days after that, he just passed on in his sleep, a healthy guy.*
(P16, Other HCW, F, Unvaccinated)

### 3.3. Confidence in Natural Immunity 

Among unvaccinated participants, there were assertions that natural immunity against COVID-19 was sufficient, deeming the vaccine unnecessary. A few participants gave accounts of previous experiences of having had COVID-19 and recovered. 


*I don’t see a need right now to get vaccinated because of the reason, like I said, my system fought it without [the vaccine].*
(P6, Other HCW, F, Unvaccinated)

Participants felt that they had already acquired natural immunity having been exposed to COVID-19. 


*People have coughed on me, they’ve sneezed on me, they’ve touched me. I know that I have got an immunity against it.*
(P24, Nurse, F, Unvaccinated)

Several participants outlined how they always relied on their natural immunity for protection against pathogens, and COVID-19 should not be treated any differently.


*And natural immunity over the years, for millennia, it’s been there, it has been shown that natural immunity… if the body’s taken care of, it can protect you against most of these pathogens. And then yeah, I’ll [opt] for natural immunity, and if I’m forced to then I’ll fight.*
(P2, Nurse, M, Unvaccinated)

### 3.4. Lack of Trust in Sources of Information 

Some hesitant participants expressed mistrust in the government and pharmaceutical companies, citing these as reasons for not getting vaccinated.


*Honestly speaking, after COVID-19 and the introduction of vaccines, it’s been difficult to trust government and difficult to trust media; it’s been difficult to trust even the World Health Organisation itself, because who are they? How did they cover some of these things, who are they working with, why is it just something blows up from China?*
(P1, Other HCW, M, Unvaccinated)

Another participant stated that although they continue to review international guidelines, they remained sceptical, particularly with regard to guidelines emanating from ‘government-sponsored organisations’. 


*Just this morning, I read some of the guidelines from the CDC and they still say it’s safe and effective, it’s better than getting COVID but I mean it’s only they will say that, and I can see with my own eyes it’s not like that. So I don’t know, I don’t trust any of these main government sponsored organisations anymore.*
(P8, Doctor, M, Unvaccinated)

Some participants felt that the vaccine was a money-making opportunity for pharmaceutical companies. 


*It was another corrupt operation this whole thing… You know, as I said, I’m 73 years old; we didn’t have this during AIDS; when the swine flu hit, we didn’t have all these, but this was turned into a corruption feast, money made, if you look at [what] Pfizer made and all these red circle vaccine companies what they have made; [they] made billions.*
(P7, Doctor, M, Unvaccinated)

### 3.5. Insufficient and Misinformation about COVID-19 Vaccines

Participants, both vaccinated and unvaccinated, felt they did not have enough information about the COVID-19 vaccines. Participants lamented the fact they could not access sufficient information about the COVID-19 vaccine and that they did not have an opportunity to ask questions about the vaccine, which seeded doubts about vaccinating.


*The only sensible information I got from colleagues why people previously infected should get the vaccine was, we think it’s better. So, there was no science behind it… I picked it up with quite a few colleagues that feel very similar, and it’s frustrating if you don’t get sensible information… If I’m not allowed to ask questions, real scientific founded questions, things that I’m worried about, that I don’t get answers to.*
(P26, Doctor, F, Unvaccinated)

This point was affirmed by another participant: 


*If you look into how the issue is handled […] not all information was given or not a clear understanding.*
(P5, Other HCW, M, Unvaccinated)

It was consistently pointed out that there was a lot of contradictory information, with one participant confessing that they initially planned to get vaccinated, but the more they read about the vaccine, the more they were convinced not to vaccinate. 


*The vaccine doesn’t work. Well, initially, I was expecting to get it, but the more I read about it, and it hasn’t been tested and it hasn’t been proven. And even up to now it’s evident that the vaccine is more harmful than useful, it’s not evading disease or getting infected or the spread of disease or complications or even getting rid of the COVID so the vaccine is known to be not helpful and it’s harmful, so that’s why I wouldn’t take it.*
(P8, Doctor, M, Unvaccinated)

## 4. Discussion

The vast majority (89%) of this sample of HCWs was vaccinated, which is above the 51% of the South African adult population vaccinated as of 12 January 2023 [30]. These findings are consistent with other studies which point to HCWs being more willing to vaccinate compared with the general public [34,35]. Vaccine uptake among HCWs is expected to be higher among HCWs because of their public health education and access to information, whilst professional societies encouraging vaccination of members may also have improved uptake rates [36]. Furthermore, HCWs faced increased exposure to COVID-19 and subsequent high levels of morbidity and mortality, thereby heightening their own perception of risk, a key factor in the decision to vaccinate [37].

Sociodemographic indicators of hesitancy mirror those found in other South African studies [19,38,39,40], which found that males and the younger cohort were less likely to vaccinate. In our study, doctors and other HCWs displayed higher levels of hesitancy compared to nurses. These study findings contrast with several international studies [6,23,26,41,42], which found that nurses were less willing to be vaccinated than other HCWs, especially physicians. HCWs working in the private sector and participants who selected African spirituality as their religion were also less likely to vaccinate, affirming other studies which illustrated religiosity as a factor driving vaccine hesitancy [43]. Vaccination may be inimical to certain religious beliefs and cultural practices, warranting further investigation in studies of factors driving vaccine hesitancy. A study undertaken among the general population in the Limpopo province in South Africa found that religious leaders can positively impact vaccination rates, but advocated for traditional leaders to be engaged if they were to play a meaningful role in improving vaccine uptake rates in rural areas [44]. Culture, whilst not a focus of this study, has been found to play a role in the uptake of COVID-19 prevention interventions. Regions exhibiting collectivist cultures were quicker to adopt nationwide mask mandates [45] or comply with lockdown regulations [46]; with African cultures considered collectivist [47], this could be a contributing factor to the high vaccination rates experienced in this study, although these high rates are not sustained amongst the general population [30].

The qualitative data provided valuable insights into the drivers of vaccine hesitancy among HCWs. This study highlighted that concerns about COVID-19 vaccine safety and effectiveness remain pervasive, even amongst those that are vaccinated. While these findings are congruent with several quantitative studies [48,49], qualitative data in this study revealed that these concerns centred around the perceived speed at which these vaccines were developed. The WHO declared a global pandemic in March 2020 [50]; by late 2020, there were already well over 200 vaccines under development and 40 vaccines in clinical trials [51]. Vaccines were also developed using mRNA technology, which is relatively new and unknown, with no previously approved mRNA vaccines despite decade-long trials [52]. The seemingly rapid availability of approved vaccines in 2021 elicited safety and efficacy concerns [12,53,54], with HCWs in this study still doubting whether the available vaccines underwent the rigorous clinical trials and regulatory reviews and approvals that they were familiar with. 

Furthermore, HCWs’ exposure to patients presenting with apparent serious adverse side-effects following vaccination, confidence in their own immune capabilities, lack of information or prevalent misinformation, and distrust of particular sources of information were noted. In previous studies, HCWs identified the perceived speed at which vaccines were developed as cause for concern [6,23,24,26,27,55]. This may intersect with misinformation or inadequate information, which feeds conspiracy theories or fuels safety fears and may ultimately fail to provide some HCWs with the confidence to get vaccinated themselves or promote vaccination to patients and the general public [56]. Even among those that were vaccinated, many felt there was inadequate information to offset their safety concerns. Despite relatively high vaccination rates among HCWs in this study, these results highlight the importance of ensuring that HCWs are meaningfully engaged and informed before and during the introduction of any new health technology, and that they have easy access to reliable sources of up-to-date vaccine-related information. Participants expressed distrust of some of the more prominent sources of information, including the WHO, government, media, and pharmaceutical companies. The distrust stemmed primarily from the mixed messages received, which can partially be attributed to the evolving nature of the epidemic, with the acquisition of new knowledge driving shifts in policy and public health directives during the pandemic. The South African Government has been criticised for their handling of the epidemic, with a general population study revealing that they were ranked low on a list of trusted sources of information [57]. Within a context of uncertainty, communication is key, something which has been found to apply not only to HCWs, but also the general public [58,59].

A review of adverse events following COVID-19 vaccination revealed that a large proportion of the small percentage of individuals who experienced adverse events presented with mild symptoms such as headaches or a fever [60]. Adverse effects such as myocarditis, glomerular diseases, and cutaneous eruptions have been associated with the mRNA vaccines but are considered rare [60]. However, due to the large number of vaccinations undertaken, HCWs are likely to encounter patients presenting with serious side-effects, thereby reflexively triggering unconscious bias. Unconscious bias is known to affect healthcare professionals in several ways, with exposure to patients presenting with adverse events potentially negatively impacting HCWs’ perception around the safety of vaccines [61]. There is also the problem of incorrect attribution where routine and unexpected clinical events are potentially attributed to vaccination history.

Some HCWs also expressed confidence in their own immune system functioning to stave off serious illness following COVID-19 infection. The superiority of natural immunity to vaccine-induced immunity is a common trope among antivaccination movements or may simply be preferential when paired with safety concerns [62,63]. The quantitative data in this study indicate that the younger cohort and those who do not suffer from any other chronic illnesses were less likely to vaccinate, and this may possibly be due to confidence in their own immune capabilities, as shown elsewhere [64]. 

Among the public, HCWs play a key role in the success of COVID-19 vaccination programmes [3,4,5,6,65], specifically because they are considered to be the most trusted source of information [66,67]. It, therefore, remains crucial that HCWs understand the value of vaccines and have the confidence to be able to communicate effectively about the merits and limitations of vaccines. If HCWs are to play a key role in promoting vaccination to the public, their own concerns will need to be allayed, their knowledge levels improved, and they will need the skills to effectively communicate to the public. To do so, they will need clear and authoritative information about vaccine development pathways, regulatory and safety oversight procedures and standards, efficacy data, common and rare side-effects, and realistic effectiveness targets (e.g., prevention of death and severe disease vs. prevention of infection). This recommendation extends beyond improving COVID-19 vaccination rates to the broader improvement of immunisation rates. Whilst multiple strategies are required, key is recognising the important role of HCWs in the vaccination process and ensuring that they are adequately capacitated to both motivate for and address public concerns with scientifically accurate and clinically relevant information [68]. 

### Strengths and Limitations

This study represents the largest cohort of HCWs in Africa to date and was undertaken 12 months after the local availability of COVID-19 vaccines. The study is also the first national study to examine COVID-19 vaccine uptake and drivers of hesitancy amongst HCWs in South Africa. A further strength is that the study employed both qualitative and quantitative approaches to better understand COVID-19 vaccine hesitancy among HCWs in South Africa. 

The study was limited by the use of an unrestricted self-administered survey that was dependent on the online reachability of HCWs on selected databases. The limitations in study design may have introduced selection bias and may have limited generalisability. High vaccination rates may have been influenced by the imposition of mandatory vaccination by certain organisations, although mandatory vaccination was not adopted by the public health sector or the larger private sector groups. Furthermore, it is unclear whether the HCWs in this study were fully vaccinated, although the Johnson and Johnson vaccine (single dose vaccine) was the only vaccine initially available in South Africa with half a million health workers accessing this vaccine through the Sisonke implementation study for HCWs [69]. 

## 5. Conclusions

Encouragingly, the majority (89%) of HCWs were vaccinated, with a relatively low (11%) proportion of the study population remaining hesitant. This study provides evidence elucidating factors that drive hesitancy, whilst also providing discursive narratives around the dominant concerns that HCWs had about COVID-19 vaccines. Concerns were fuelled by the perceived speed at which these vaccines were developed, together with the lack of adequate information, mixed messages emanating from government and international organisations, and prevalent misinformation accessed by HCWs. Unvaccinated HCWs also placed faith in their own immune system’s ability to stave of serious illness, while they raised concern about the perceived high rates of serious adverse events. These issues have individually and collectively sowed seeds of doubt around the safety and effectiveness of the available COVID-19 vaccines. These concerns require addressing if HCWs are to acquire the requisite confidence in the vaccines, which would improve and sustain vaccine and booster uptake rates within this critical subpopulation. Broad uptake of effective COVID-19 vaccines will be essential to reducing hospitalisations, deaths, and possibly COVID-19 infection. HCWs can be important advocates for COVID-19 vaccines, making it critical for this workforce to be well trained if they are going to play a meaningful role in improving the uptake of vaccines among the general population in South Africa, which currently remains suboptimal.

## Figures and Tables

**Table 1 vaccines-11-00414-t001:** Descriptive information and characteristics of healthcare workers according to vaccination status.

		Unvaccinated (Base Case)	Vaccinated	Odds Ratio [95% C. I.]
Measures	Total participants n (%)	n (%)	n (%)	
Age (*p*-value = 0.006) ^2^				
Younger than 35 years old	2259 (31.83)	266 (12.09)	1934 (87.91)	1.00
35 to 49 years old	2906 (40.95)	273 (9.62)	2565 (90.38)	1.29 [1.08–1.54]
50 years old or older	1932 (27.22)	180 (9.46)	1723 (90.54)	1.31 [1.07–1.60]
Gender (*p*-value = 0.013)				
Male	2168 (30.66)	251 (11.78)	1880 (88.22)	1.00
Female	4904 (69.34)	469 (9.80)	4317 (90.20)	1.22 [1.04–1.44]
Race (*p*-value = 0.000)				
Black African	4042 (58.33)	392 (9.96)	3542 (90.04)	1.00
Coloured	527 (7.61)	60 (11.86)	446 (88.14)	0.82 [0.61–1.09]
Indian	427 (6.16)	24 (5.67)	399 (94.33)	1.83 [1.20–2.81]
White	1933 (27.90)	226 (11.79)	1691 (88.21)	0.82 [0.69–0.98]
Religion (*p*-value = 0.000)				
Christian	5668 (82.68)	568 (10.25)	4973 (89.75)	1.00
Muslim	295 (4.30)	28 (9.69)	261 (90.31)	1.06 [0.71–1.58]
Buddhist or Hindu	228 (3.33)	11 (4.85)	216 (95.15)	2.24 [1.21–4.13]
African Spirituality	194 (2.83)	34 (17.99)	155 (82.01)	0.52 [0.35–0.76]
Other	470 (6.86)	50 (10.85)	411 (89.15)	0.93 [0.69–1.27]
Nationality (*p*-value = 0.627)				
South African	6233 (89)	630 (10.33)	5470 (89.67)	1.00
Non-South African	770 (11)	82 (10.90)	670 (89.10)	0.94 [0.73–1.20]
Occupation (*p*-value = 0.000)				
Nurse	2568 (44.45)	184 (7.30)	2337 (92.70)	1.00
Doctor	1169 (20.24)	108 (9.33)	1049 (90.67)	0.76 [0.59–0.98]
All other	2040 (35.31)	219 (11.02)	1768 (88.98)	0.63 [0.51–0.78]
Sector (*p*-value = 0.002)				
Public	2353 (40.72)	196 (8.48)	2114 (91.52)	1.00
Private	2069 (35.81)	214 (10.58)	1809 (89.42)	0.78 [0.63–0.96]
NGO	555 (9.61)	30 (5.48)	517 (94.52)	1.59 [1.07–2.37]
Public and private	507 (8.77)	45 (9.05)	452 (90.95)	0.93 [0.66–1.30]
Other	294 (5.09)	26 (9.00)	263 (91.00)	0.93 [0.61–1.43]
Years worked (*p*-value = 0.718)				
Less than 5 years	811 (14.04)	72 (9.08)	721 (90.92)	1.00
5 to 9 years	1310 (22.68)	123 (9.57)	1162 (90.43)	0.94 [0.69–1.28]
10 years or more	3655 (63.28)	316 (8.81)	3270 (91.19)	1.03 [0.79–1.35]
Chronic conditions ^1^ (*p*-value = 0.000)				
No	4633 (59.68)	521 (11.50)	4009 (88.50)	1.00
Yes	3130 (40.32)	202 (8.32)	2226 (91.68)	1.43 [1.20–1.69]
COVID-19 history (*p*-value = 0.121)				
No	3589 (50.46)	382 (10.95)	3108 (89.05)	1.00
Yes	3524 (49.54)	340 (9.81)	3125 (90.19)	1.12 [0.96–1.31]

^1^ Chronic conditions were defined as having one or more of the following: diabetes, hypertension, respiratory disease, HIV, and other chronic diseases. ^2^ Model significance from univariate logistic regression.

## Data Availability

The data presented in this study are available on request from the corresponding author.

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
