# Peer review of "Understanding COVID-19 Vaccine Hesitancy among Healthcare Workers in South Africa"

_vaccines, 2023, doi:10.3390/vaccines11020414_

Round 1

Reviewer 1 Report

Abstract

line 20: 89% had been vaccinated

Introduction

Clear and informative. Included information from a global perspective.

Clear aims

Methods and Results 

Thorough and clear. Appropriate statistical methods. Good sample size

You could break up some larger blocks of text 

Discussion points - to take or leave 

Thorough cover of literature. 

With regards to 11% of HCWs showing vaccine hesitancy or confidence in resilience of their immune system:  the young age profile of the South African population,  a climate that mitigates against the spread of COVID due to the amount of outdoor living and the cultural respect for traditional medicine perhaps deserve  more emphasis? The Covid  pandemic had mercifully not been so pervasive in Africa because of climatic cultural and age profiles in many African countries. At a Zoom webinar run by the Economist magazine on 14 January 2021 Covid vaccines your questions answered' John McDermott the Economist correspondent for Africa gave very well informed views on these advantages.   Africa correspondent @The Economist 

11% is surely  a laudably low figure for vaccine hesitancy and points to the  success of the education and information programmes about vaccine development, in South Africa. This is  particularly true  it would seem, among  those working in the public sector.  Can low  vaccine hesitancy in public service HCWs be used as a  proxy for the standard of education and calibre of the disseminated information?  South Africa has a well established  mobile phone-owning public and mobile phone network. Presumably, the bulk of information was disseminated in this way and  calibre of  the information would have more influence on attitudes to vaccine development and uptake than any barriers to dissemination.

Concerns about  the rapidity of the covid vaccine trial and the subsequent rapidity of the vaccine roll out  compared with  previous vaccine trial design experience were described as major reasons for vaccine hesitancy in HCWs, both vaccinated and not vaccinated.  This was  a major point of misunderstanding among HCWs globally,  as the reference list in this study as well as the results in this study reveal.

I think  this important point needs drawing out - you seem to have got to the root of the vaccination hesitancy problem with your large sample size and people in both the vaccination and non vaccination camps expressing concerns on this issue. It seemed to lie at the root of many of the questionnaire responses exampled. Was WHO messaging was very inadequate here?

Is it worth emphasising  that the trial design for vaccine efficacy in covid  was unique, and could not be compared with previous trials.  Instantaneous replication of data in trials conducted on a pan-global scale of replication, for spike protein manipulation-type vaccines and MRNA-type vaccines simultaneously,  had not been done before. The data on vaccine efficacy thus  arrived very quickly within months instead of years and acceptable statistical power for confidence in rapid vaccine roll out arrived correspondingly quickly as well.  Maybe this aspect alone was what lay behind the bulk of safety and efficacy concerns, leading to vaccine hesitancy in some.

There was also a global lack of appreciation  of the head start  the Astra Zeneca vaccine had through MERS virus work in the early 2000s (https://www.nature.com/articles/d41586-020-03626-1). The MRNA Pfizer/BioNTech  work stemmed from the imaginative brains of UÄŸur Åžahin, a German immunologist who applied a solution from an oncological to a novel context. 

Author Response

Reviewer’s Comment 

Response

Location of Response in Revised Manuscript

Reviewer 1

With regards to 11% of HCWs showing vaccine hesitancy or confidence in resilience of their immune system:  the young age profile of the South African population,  a climate that mitigates against the spread of COVID due to the amount of outdoor living and the cultural respect for traditional medicine perhaps deserve  more emphasis? The Covid  pandemic had mercifully not been so pervasive in Africa because of climatic cultural and age profiles in many African countries. At a Zoom webinar run by the Economist magazine on 14 January 2021 Covid vaccines your questions answered' John McDermott the Economist correspondent for Africa gave very well informed views on these advantages.   Africa correspondent @The Economist 

The reviewer has raised an important point that we have considered and included in the manuscript. Research has established that Africa has had the slowest COVID-19 infection rates and low severity of illness. This has been mainly attributed to the Africa’s young, rural population, restrictions/lockdowns, and arguably lower rates of international travel (compared to other continents). Conversely, testing and reporting have also been problematic, potentially masking the true extent of infection rates and attributable morbidity and mortality.

We have included this point in the introduction section in the manuscript, providing additional context for the potential reasons for vaccine hesitancy.

Introduction, line 63-73

11% is surely  a laudably low figure for vaccine hesitancy and points to the  success of the education and information programmes about vaccine development, in South Africa. This is  particularly true  it would seem, among  those working in the public sector.  Can low  vaccine hesitancy in public service HCWs be used as a  proxy for the standard of education and calibre of the disseminated information?  South Africa has a well-established  mobile phone-owning public and mobile phone network. Presumably, the bulk of information was disseminated in this way and  calibre of  the information would have more influence on attitudes to vaccine development and uptake than any barriers to dissemination.

The reviewer has raised a critical point which needs attention in the manuscript. We have now highlighted that HCW’s are more educated in public health matters, with more exposure to medical information than the general public. Furthermore, it can be argued that HCWs are at more risk of infection and have observed more COVID-19 related morbidity and mortality than the general public. These are some of the factors that are highlighted in previous studies in other parts of the world.

Discussion, line 297 to 305

Concerns about  the rapidity of the covid vaccine trial and the subsequent rapidity of the vaccine roll out  compared with  previous vaccine trial design experience were described as major reasons for vaccine hesitancy in HCWs, both vaccinated and not vaccinated.  This was  a major point of misunderstanding among HCWs globally,  as the reference list in this study as well as the results in this study reveal.

I think  this important point needs drawing out - you seem to have got to the root of the vaccination hesitancy problem with your large sample size and people in both the vaccination and non-vaccination camps expressing concerns on this issue. It seemed to lie at the root of many of the questionnaire responses exampled. Was WHO messaging was very inadequate here?

Is it worth emphasising  that the trial design for vaccine efficacy in covid  was unique, and could not be compared with previous trials.  Instantaneous replication of data in trials conducted on a pan-global scale of replication, for spike protein manipulation-type vaccines and MRNA-type vaccines simultaneously,  had not been done before. The data on vaccine efficacy thus  arrived very quickly within months instead of years and acceptable statistical power for confidence in rapid vaccine roll out arrived correspondingly quickly as well.  Maybe this aspect alone was what lay behind the bulk of safety and efficacy concerns, leading to vaccine hesitancy in some.

There was also a global lack of appreciation  of the head start  the Astra Zeneca vaccine had through MERS virus work in the early 2000s (https://www.nature.com/articles/d41586-020-03626-1). The MRNA Pfizer/BioNTech  work stemmed from the imaginative brains of UÄŸur Åžahin, a German immunologist who applied a solution from an oncological to a novel context. 

The perceived speed at which the COVID-19 vaccines were developed was a concern among unvaccinated and vaccinated HCWs. 

As the reviewer has rightfully pointed out, these concerns were felt globally among HCWs with particular concern around the mRNA technology. In the discussion session we have teased this point out further by highlighting how this new technology and the speed at which approved vaccines were made available may have contributed to the uncertainty indicated by some of the study participants.

We did not expand on the merits of the technology or focus on specific vaccines as we felt this was beyond the scope of this paper.

Discussion, line 322 to 334

line 20: 89% had been vaccinated

Thank you; this grammatical error was revised accordingly.

Abstract, line 21

Reviewer 2 Report

Many thanks for giving me the opportunity to read this paper. I believe that the study is interesting and covers a whole range of questions related to vaccinations, e.g. information from health care workers (HCW) to patients, and how to convince HCW themselves to get vaccinated. This piece of work is interesting and should be considered for publication in Vaccine. I however have several suggestions upon publication:

1- Cultural context. South Africa is characterized by different ethnies (accounted for in the publication) but little is known about how different ethnies perceive vaccination. However, vaccine hesitancy is higher when religious or beliefs go against vaccination. I believe that cultural differences (e.g. as discusses by Chen, Frey & Presidente 2021 or An, Porcher, Tang & Kim 2021 and Porcher 2021) should be discussed. Different culture lead to different behavior and different outcomes. 

2- Political differences. I am wondering whether polarization in political terms can have an impact and whether you could discuss it. An important study in the US is by Porcher and Renault (2021) in the US who show that democrats are more respectful of social distancing than republicans. I am wondering whether these results could make sense in the South African context. HCW might be closer to democrats as they work in social services and might be more encline to follow the precepts from the government.

3. Finally, what was the situation in terms of government ridigity at the time of the study ? Where we still in lockdown or restrictions in South Africa ?

Author Response

Reviewer 2

1- Cultural context. South Africa is characterized by different ethnies (accounted for in the publication) but little is known about how different ethnies perceive vaccination. However, vaccine hesitancy is higher when religious or beliefs go against vaccination. I believe that cultural differences (e.g. as discusses by Chen, Frey & Presidente 2021 or An, Porcher, Tang & Kim 2021 and Porcher 2021) should be discussed. Different culture lead to different behavior and different outcomes. 

The reviewer has made a plausible and valuable point which we have considered, however, our data presents demographics which cover religion and not culture. We see no argument worth raising about South Africa’s different ethnic groups because we do not have data to support this assertion.

We agree that cultural factors could play a critical role in vaccine uptake and policy implementation, but this this factor was not included in our research tools.

We have noted that HCWs who selected African spirituality as their religion were less likely to vaccinate, but this is within a context of high HCW vaccination rates. We have further noted that other studies have illustrated religiosity as a factor driving vaccine hesitancy.

N/A

2- Political differences. I am wondering whether polarization in political terms can have an impact and whether you could discuss it. An important study in the US is by Porcher and Renault (2021) in the US who show that democrats are more respectful of social distancing than republicans. I am wondering whether these results could make sense in the South African context. HCW might be closer to democrats as they work in social services and might be more incline to follow the precepts from the government.

While in most European and American contexts, political differences may contribute to vaccine acceptance and uptake, South Africa is characterized by a plethora of political parties, and whilst separated by ideology, did not take overtly contrasting stances on the issue of vaccination.   

Political differences were not assessed in our survey.

N/A

3. Finally, what was the situation in terms of government rigidity at the time of the study? Where we still in lockdown or restrictions in South Africa?

At the time of the study, which was between August 2022 and October 2022, national lockdowns and restrictions were minimal with most public spaces accessible. With vaccines being available to everyone, the government in South Africa had little rigidity even around domestic travel.

N\A

Reviewer 3 Report

It is a very interesting study which aimed to assess willingness to accept vaccination against COVID-19 among healthcare workers in South Africa and the reasons of their hesitancy. It is important to understand HCWs concerns because their vaccination attitudes predict their level of vaccination uptake and intention to recommend vaccinations not only against covid-19, but generally, to their patients.

Author Response

Reviewer 3

It is a very interesting study which aimed to assess willingness to accept vaccination against COVID-19 among healthcare workers in South Africa and the reasons of their hesitancy. It is important to understand HCWs concerns because their vaccination attitudes predict their level of vaccination uptake and intention to recommend vaccinations not only against covid-19, but generally, to their patients.

Noted and well received feedback.

N/A

Round 2

Reviewer 2 Report

The authors respond to my comments in the letter but not in the text...cultural differences, political differences, etc. even if not assessed should be put in limitations & references to the literature on these issues should be integrated.

Author Response

We have reconsidered and made substantive mention of the role of religion and culture in adopting COVID-19 prevention measures-  including vaccination. See lines 314 to 325 in the manuscript. 

Round 3

Reviewer 2 Report

It can be accepted in present form I believe.